# Construction of 3D copper-chitosan-gas diffusion layer electrode for highly efficient $CO_2$ electrolysis to $C_{2+}$ alcohols

Jiahui Bi[1,2], Pengsong Li[1,2], Jiyuan Liu[1,2], Shuaiqiang Jia[3], Yong Wang[1,2], Qinggong Zhu [1,2] ✉, Zhimin Liu [1,2] & Buxing Han [1,2,3] ✉

High-rate electrolysis of $CO_2$ to $C_{2+}$ alcohols is of particular interest, but the performance remains far from the desired values to be economically feasible. Coupling gas diffusion electrode (GDE) and 3D nanostructured catalysts may improve the efficiency in a flow cell of $CO_2$ electrolysis. Herein, we propose a route to prepare 3D Cu-chitosan (CS)-GDL electrode. The CS acts as a "transition layer" between Cu catalyst and the GDL. The highly interconnected network induces growth of 3D Cu film, and the as-prepared integrated structure facilitates rapid electrons transport and mitigates mass diffusion limitations in the electrolysis. At optimum conditions, the $C_{2+}$ Faradaic efficiency (FE) can reach 88.2% with a current density (geometrically normalized) as high as 900 mA cm$^{-2}$ at the potential of −0.87 V vs. reversible hydrogen electrode (RHE), of which the $C_{2+}$ alcohols selectivity is 51.4% with a partial current density of 462.6 mA cm$^{-2}$, which is very efficient for $C_{2+}$ alcohols production. Experimental and theoretical study indicates that CS induces growth of 3D hexagonal prismatic Cu microrods with abundant Cu (111)/Cu (200) crystal faces, which are favorable for the alcohol pathway. Our work represents a novel example to design efficient GDEs for electrocatalytic $CO_2$ reduction ($CO_2$RR).

The electrocatalytic $CO_2$ reduction reaction ($CO_2$RR) is of great significance for reducing the consumption of fossil resources and close the carbon-neutral energy cycle[1–8]. The multi-carbon ($C_{2+}$) products, especially $C_{2+}$ alcohols, are of particular interest in many applications. To date, the most efficient system to produce $C_{2+}$ products is using flow cell assembly[9–11]. In this configuration, $CO_2$ electrolysis occurs at the gas–liquid–solid three-phase interface, and the $CO_2$RR activity is often limited by the size of the interface area and mass transport. For the catalyst layer, the most straightforward way is a coating of powder-type electrocatalysts onto a gas diffusion layer (GDL) using commonly used polymers/binders, such as polyaniline (PANI), polypyrrole (PPy),

and Nafion D-521[12–14]. However, the additive binders would inevitably decrease the $CO_2$RR performance and considerably increase the overpotential, which is due to the obstruction of gas transport, insufficient exposure of active sites, and detachment of catalyst from electrode surface by binder degradation in the reaction[15–17]. To this end, the design of the gas diffusion electrode (GDE), which consists of the catalyst layer and GDL, mainly focuses on two aspects. One is to expand the reaction interface to improve the utilization of electrocatalysts. The other is to construct efficient electrons, $CO_2$, and product transport networks to reduce the ohmic and mass transport losses of $CO_2$RR.

[1]Beijing National Laboratory for Molecular Sciences, CAS Key Laboratory of Colloid, Interface and Chemical Thermodynamics, CAS Research/Education Center for Excellence in Molecular Sciences, Center for Carbon Neutral Chemistry, Institute of Chemistry, Chinese Academy of Sciences, 100190 Beijing, P. R. China. [2]School of Chemistry and Chemical Engineering, University of Chinese Academy of Sciences, 100049 Beijing, P. R. China. [3]Shanghai Key Laboratory of Green Chemistry and Chemical Processes, School of Chemistry and Molecular Engineering, East China Normal University Shanghai, 200062 Shanghai, P. R. China. ✉e-mail: qgzhu@iccas.ac.cn; hanbx@iccas.ac.cn

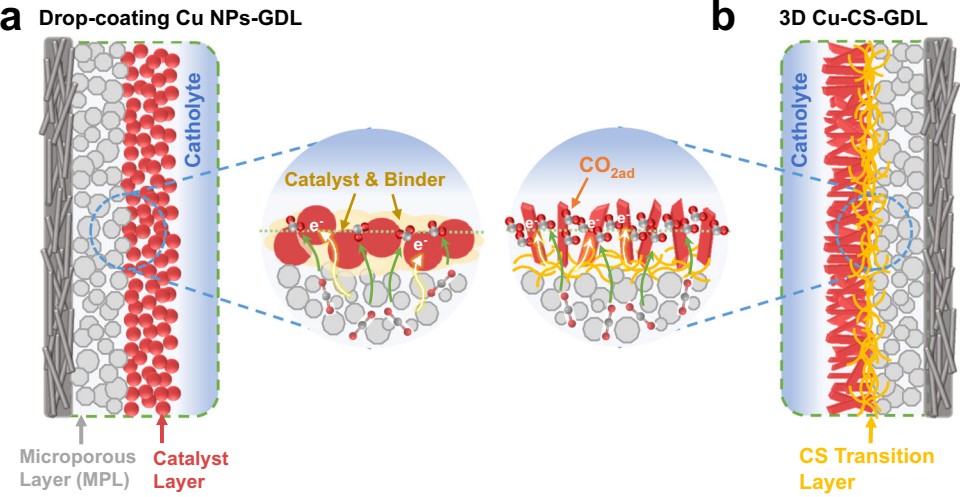

**Fig. 1 | Schematic illustration of the conventional and the as-synthesized 3D Cu–CS-GDL GDEs in a flow cell. a** Structure of the GDE prepared via the conventional drop-coating method. **b** 3D Cu–CS-GDL electrode prepared in this study.

One of the most important issues to enhance $CO_2RR$ efficiency is the development of GDEs that facilitate rapid electron transport and mitigate mass diffusion limitations. This requires catalysts that can expose abundant active sites and rapidly deliver $CO_2$ and electrons to the catalysts[18–22]. A few pioneer works have shown that the catalytic performance of $CO_2RR$ can be improved by growing 2D catalyst thin-film layers on the surface of the GDL. Remarkably, an integrated N-C/Cu/PTFE architecture that has a strong electron-donating ability and confinement of the nitrogen-doped carbon layers enabled advances in selectivity towards ethanol with a Faradaic efficiency (FE) of 52% and a partial current density of 156 mA cm$^{-2}$[23]. Over integrated Ce(OH)$_x$/Cu/PTFE GDE, the FE of ethanol could reach 43% with a partial current density of 128 mA cm$^{-2}$[24]. The 2D structure enables a uniform $CO_2$ reactant concentration and local reaction environment throughout the catalyst layer, which can enhance both the activity and selectivity of $C_{2+}$ alcohols. Compared with 2D structures, 3D nanostructured catalysts that exhibit interesting morphological properties have been demonstrated as efficient catalysts for $CO_2RR$. While an electrodeposition process can also create a 3D material, it is difficult to deposit a 3D catalyst directly on a GDL due to the hydrophobic surface of the GDL, which would lose its gas diffusion function and cause flooding during $CO_2RR$. The other is low efficient use of the total catalyst active sites.

Chitosan (CS), an abundant amino polysaccharide, obtained from the carapaces of shrimp and crabs, containing a carbon skeleton with amino-functional groups[25–27]. It has the advantages of low cost, non-toxic, renewable, degradable, and abundant reserves, which has some unique advantages comparing with commonly used polymers/binders. The hydroxyl group and amino group in the CS structure make it a strong affinity, especially has good chelation ability for transition metals, coordinating with metal ions to form complexes, this property also provides a basis for dispersing metal active sites[25–28]. In addition, CS has been proven to have the ability of structure guidance and good adsorption of $CO_2$[29,30]. These features of CS made it an interesting material in designing electrocatalysts for $CO_2RR$.

Herein, we propose a method to prepare a 3D Cu–CS-GDL electrode. The CS can act as a "transition layer" between the catalyst and GDL. By systematically tailoring the transition layer, $CO_2$ and charge transport in the electrode are optimized to obtain high $C_{2+}$ alcohol productivity. The $C_{2+}$ FE could reach 88.2% with a current density (geometrically normalized) as high as 900 mA cm$^{-2}$ at the potential of −0.87 V vs. reversible hydrogen electrode (RHE), of which the $C_{2+}$ alcohols selectivity was 51.4% with a partial current density of

462.6 mA cm$^{-2}$. It was also found that CS-induced growth of 3D hexagonal prismatic Cu microrods with abundant Cu (111)/Cu (200) crystal faces, which were favorable for the alcohol pathway. Moreover, the 3D Cu–CS-GDL structure also facilitated rapid electron transport and mitigated $CO_2$ diffusion limitations in the reaction.

## Results

The structures of the GDEs prepared by the conventional drop-coating method and our method are shown in Fig. 1. The drop-coating Cu nanoparticles (NPs)-GDL electrode has the disadvantage that $CO_2$ gas transport is obstructed and the active sites of the catalyst expose insufficiently, displayed in Fig. 1a, leading to low reaction rate and low FE in $CO_2RR$. By contrast, $CO_2$ and charge can transport quickly in a 3D Cu–CS-GDL electrode, profiting from the arrangement of the catalysts perpendicular to GDL (Fig. 1b), which results in obtaining high $C_{2+}$ alcohol productivity with high-efficiency electrolysis.

Figure 2a showed the synthesis process of 3D Cu–CS-GDL electrode. In this route, the hydroxyl group and amino group of the CS structure (Supplementary Fig. S1) make it possess a strong chelating ability, which can directly coordinate with Cu$^{2+}$ to obtain Cu–CS complexes[28]. The complexes were then drop-coated on the polytetrafluoroethylene (PTFE)-hydrophobized carbon paper (CP) and Cu$^{2+}$ was in situ reduced to Cu NPs electrochemically to obtain pre-Cu–CS-1. Then more Cu was loaded by an electrodeposition process using the Cu particles in pre-Cu–CS-1 as the nuclei and pre-Cu–CS-2 was obtained, in which Cu$_2$O and Cu coexisted. Finally, Cu$_2$O in the pre-Cu–CS-2 was in situ reduced to metallic Cu under the same operating condition of the $CO_2RR$, and 3D hexagonal prismatic Cu microrods could be formed after 10 min of $CO_2RR$ in a flow cell due to the reconstruction of Cu atoms during the change from Cu (I) to Cu (0).

Initially, the pre-Cu–CS-1 catalyst displayed a uniform dispersion of Cu NPs, which was confirmed by transmission electron microscopy (TEM) characterization. The size of Cu NPs was further grown in pre-Cu–CS-2 catalyst, suggesting a substantial enrichment of Cu atoms (Supplementary Fig. S2). Deriving from the pre-Cu–CS-2 catalyst, a 3D oriented hexagonal prismatic Cu microrods gradually formed on the CS transition layer in 10 min of situ reduction (Supplementary Fig. S3, Fig. 2b). The contact angle of 3D Cu–CS-GDL was 95.1° (Fig. 2b-inset), suggesting that CS could act as a "transition layer" between catalyst and microporous layer, which not only stabilized the hydrophobic surface of the GDL to a certain extent, but also inducing growing of oriented Cu films (Fig. 2c). The presence of CS transition layer was further confirmed by Fourier transform infrared (FT-IR) results

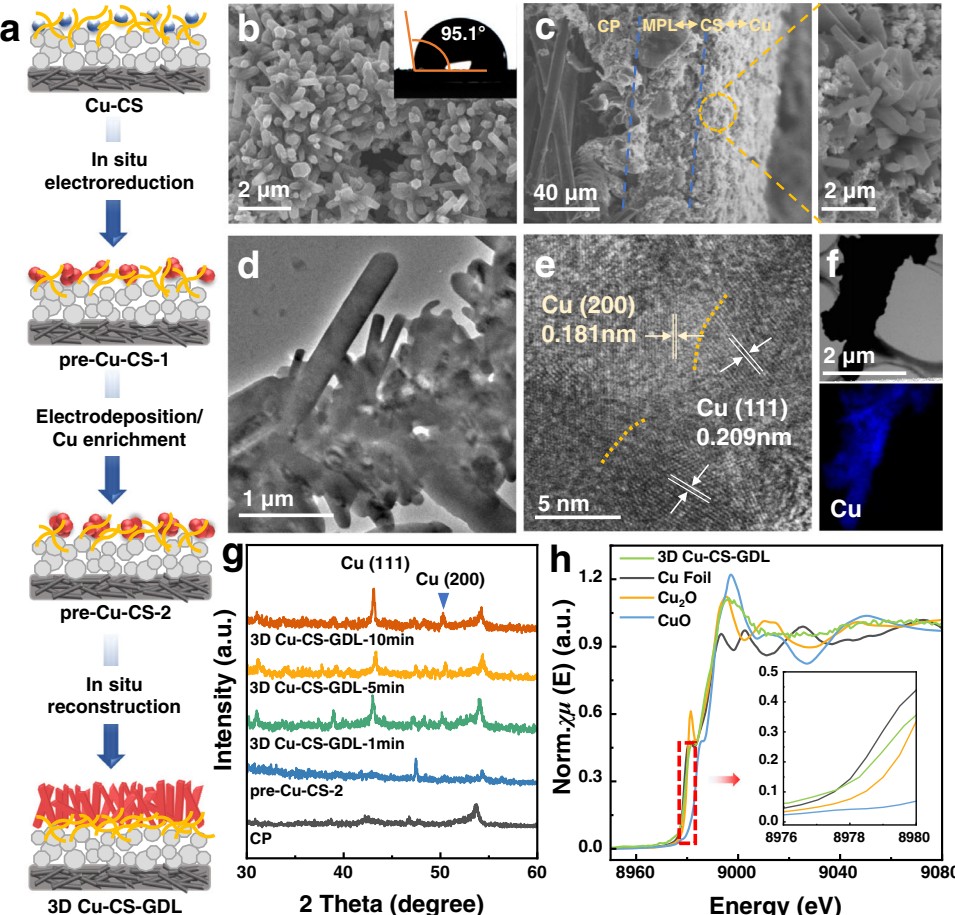

**Fig. 2 | Characterizations of 3D Cu–CS-GDL electrode. a** Schematic diagram of the process for preparation of the 3D Cu–CS-GDL electrode. **b** The SEM image and contact angle data of 3D Cu–CS-GDL electrode. **c** The side view of the SEM image of 3D Cu–CS-GDL electrode. **d** The TEM image of 3D Cu–CS-GDL electrode. **e** The HRTEM image of 3D Cu–CS-GDL electrode. **f** The EDS images of 3D Cu–CS-GDL electrode. **g** The XRD patterns of CP, 3D Cu–CS-GDL electrode with different in situ electroreduction times. **h** The normalized Cu K edge XANES spectra of 3D Cu–CS-GDL electrode.

(Supplementary Fig. S4), and the side-view energy dispersive X-ray spectroscopy (EDS) mapping of the GDE also indicated that the CS was mainly distributed between Cu and GDL (Supplementary Fig. S5). In addition, the TEM image in Fig. 2d also demonstrated the microrod structure of Cu in 3D Cu–CS–GDL. High-resolution transmission electron microscopy (HRTEM) image further showed that the lattice spacing of Cu was 0.181 nm and 0.209 nm, corresponding to the lattice plane distance of Cu (200) and Cu (111) in Cu microrods (Fig. 2e). The EDS analysis further confirmed the homogeneous distribution of Cu species (Fig. 2f).

The X-ray powder diffraction (XRD) patterns of 3D Cu–CS-GDL are shown in Fig. 2g. The peaks located at 43.3° and 50.4° can be indexed to the (111) and (200) crystalline planes of metallic Cu. Furthermore, Cu K-edge X-ray absorption near edge structure (XANES) (Fig. 2h) clearly revealed that the Cu K edge position (8979.16 eV) of 3D Cu–CS-GDL was similar to that of metallic Cu (8980.28 eV), which also indicated that mainly metallic Cu species existed in 3D Cu–CS-GDL. The semi-in-situ X-ray photoelectron spectroscopy (XPS) (Supplementary Fig. S6) spectra of Cu 2$p$ of 3D Cu–CS-GDL also indicated that Cu mainly existed in the metallic state. Notably, the decreasing signal of N 1$s$ ascribed to the coverage from the upper Cu layer, indicating that CS was below the Cu catalyst layer.

To verify the advantage of the 3D architecture, we also prepared a series of Cu GDEs for comparison, including (1) Cu NPs drop-coating onto GDL (Cu NPs-GDL), (2) Cu/CS composite drop-coating onto GDL (Cu/CS-GDL), and (3) electrodeposited Cu onto GDL (De–Cu-GDL). For

Cu NPs-GDL electrode, Cu NPs[31] was prepared by hydrothermal method and drop-coated onto the GDL using a binder (Supplementary Fig. S7). The average particle size of Cu NPs was 25 nm and possessed a metallic Cu crystal structure with exposed Cu (111) crystal planes (Supplementary Figs. S8–S10). For Cu/CS-GDL electrode, we synthesized it using a one-pot method (Supplementary Fig. S11). The Cu/CS composite was made of Cu NPs and CS. Supplementary Fig. S12a–e shows the as-synthesized Cu/CS composite before and after 10 min of CO$_2$RR, indicating that the bulk structure of the catalyst was stable during the reaction. Supplementary Fig. S12f and g further illustrated that Cu NPs with an average particle size of around 5 nm were uniformly distributed in the CS network structure. The composite also contained metallic Cu (111) crystal structure and Cu$^0$ as the main species (Supplementary Figs. S12h, S13, and S14). We also prepared electrodeposited Cu onto GDL without using CS (De–Cu-GDL, Supplementary Fig. S15). The De–Cu-GDL clearly exhibited a 3D dendrite structure, which mainly included metallic Cu (111) and Cu (200) crystal structures (Supplementary Figs. S16–S18). Unfortunately, direct electrodeposition of a 3D De–Cu on a GDL significantly destroyed the hydrophobic surface of the layer (70° contact angle) because the Cu blocked some pores of GDL, and thus the gas diffusion function of the layer was reduced during CO$_2$RR.

The catalytic activity of the 3D Cu–CS-GDL electrode for CO$_2$RR was studied by linear-sweep voltammetry (LSV) curves in a flow cell reactor (Supplementary Fig. S19), in which the electrolyte was 1 M KOH aqueous solution, and the LSV curves are shown in Supplementary

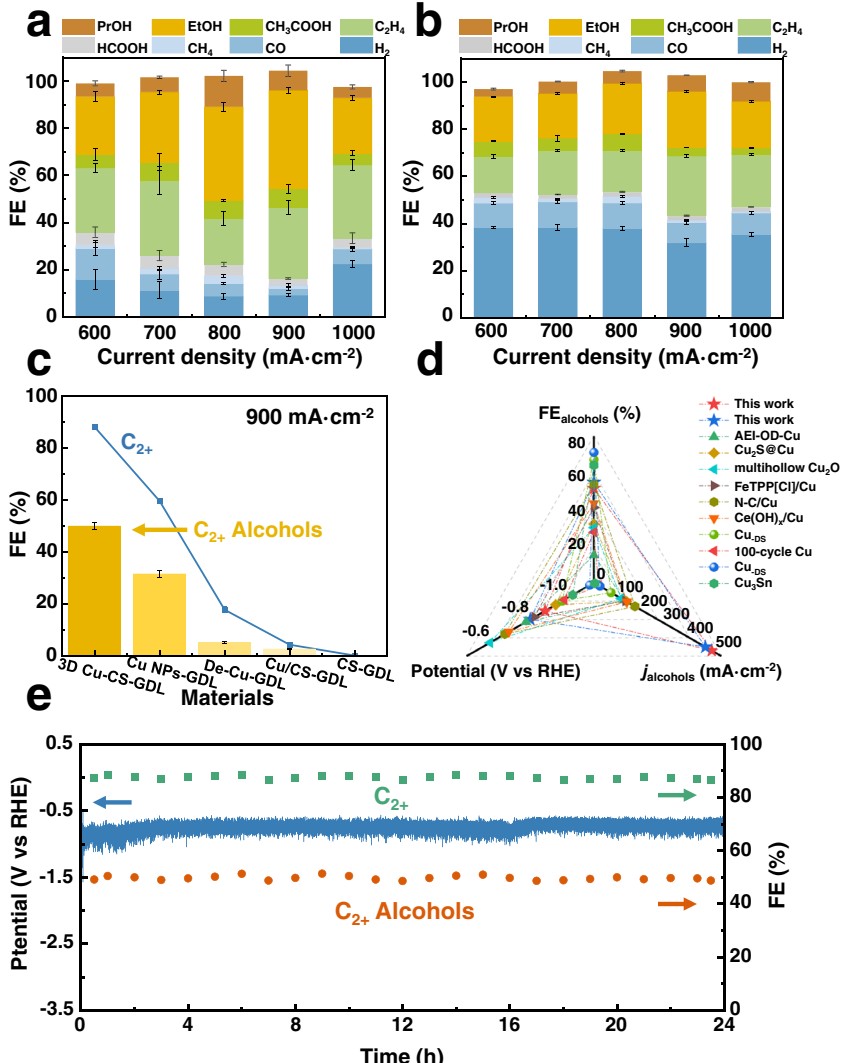

**Fig. 3 | Performance of CO₂RR in a flow cell.** Product distributions of CO₂RR on **a** 3D Cu–CS-GDL and **b** Cu NPs-GDL electrodes ranging from 600 to 1000 mA cm⁻². **c** The FE of C₂₊ products and C₂₊ alcohols on different GDEs at a current density of 900 mA cm⁻². Error bars denote the standard deviations from multiple measurements. **d** Comparison of FE of C₂₊ alcohols, the partial current density of C₂₊ alcohols ($j_{alcohols}$) and applied potential over 3D Cu–CS-GDL electrode and typical Cu-based catalysts reported. Detailed data are shown in Table S1. **e** Long-term stability of 3D Cu–CS-GDL electrode at 900 mA cm⁻².

Fig. S20. The comparison between Cu NPs-GDL and Cu/CS-GDL electrodes illustrated that the addition of CS was beneficial to increase the current density at the same reaction potential. The current density of the 3D Cu–CS-GDL electrode was higher than that of Cu NPs-GDL, Cu/CS-GDL, and De–Cu-GDL electrodes at the potential range from −0.1 to −0.6 V vs. RHE.

The electrochemical CO₂RR performance of the as-prepared catalysts was also investigated by electrolysis of CO₂ at different applied current densities ranging from 600 to 1000 mA cm⁻². The gaseous products H₂, CO, CH₄, and C₂H₄ were determined by gas chromatography (GC), and the liquid products HCOOH, CH₃CH₂OH (EtOH), CH₃COOH, and CH₃CH₂CH₂OH (PrOH) were detected using ¹H nuclear magnetic resonance (¹H-NMR) (Supplementary Fig. S21). The performances of 3D Cu–CS-GDL, Cu NPs-GDL, Cu/CS-GDL, and De–Cu-GDL electrodes were investigated for CO₂RR. As shown in Fig. 3a, b, Supplementary Figs. S22 and S23, all the electrodes yielded products with a combined FE of around 100%. For the 3D Cu–CS-GDL electrode, the FE of C₂₊ products was up to 88.2% with a current density as high as 900 mA cm⁻² at -0.87 V vs. RHE, in which the FE of C₂₊ alcohols could reach 51.4% with a partial current density of 462.6 mA cm⁻² (Fig. 3a). Interestingly, the highest FE of C₂₊ alcohols could reach up to 54.7%

(Supplementary Fig. S24), which was nearly 3 times of C₂H₄, indicating that 3D Cu–CS-GDL electrode is more inclined to the alcohol reaction pathway. Comparably, the maximum C₂₊ FE for the Cu NPs-GDL electrode was 59.6%, and the FE of C₂₊ alcohols was only 27.1%, which was much lower than that of the 3D Cu–CS-GDL electrode (Fig. 3b). For Cu/CS-GDL, De–Cu-GDL and CS-GDL (dropping CS onto GDL) electrodes, the C₂₊ alcohols FEs were below 10% in all the potential range (Fig. 3c, Supplementary Fig. S25). In addition, we have also used commercial Cu NPs for comparison (Supplementary Fig. S26). The real size of commercial Cu NPs was approximately 60–400 nm. As a result, the FE of C₂₊ was only 10.3% and H₂ was the major product at 900 mA cm⁻² (Supplementary Fig. S27). The above results indicate that the high activity of 3D Cu–CS-GDL electrode originated from the combined function of the 3D structure and the CS transition layer. Systematic comparisons of C₂₊ alcohol performance to state-of-the-art Cu-based electrocatalysts revealed that the as-synthesized 3D Cu–CS-GDL electrode was among the outstanding catalysts for C₂₊ products, especially for the high-rate production of C₂₊ alcohols (Fig. 3d, Supplementary Table S1)[11,23,24,32–37].

Considering the experimental observations above, we thought that the 3D architecture could create a favorable microenvironment to

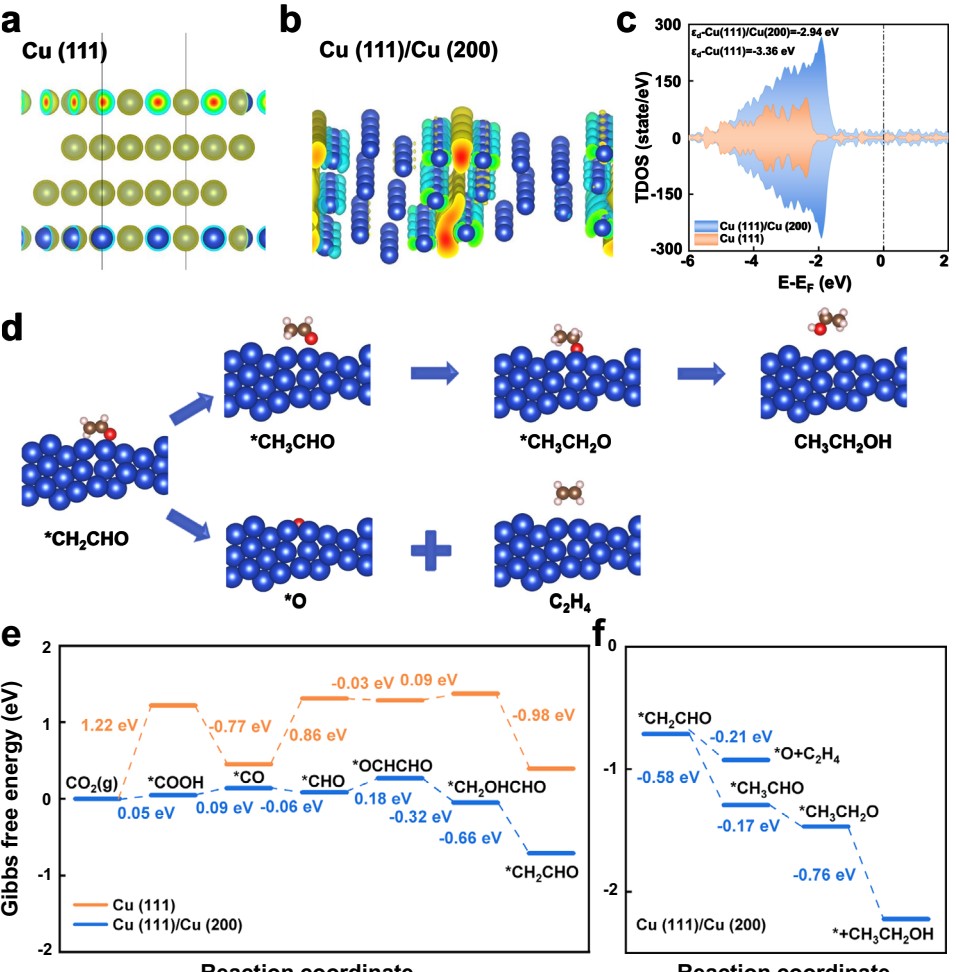

**Fig. 4 | Theoretical studies.** The model and the charge difference diagrams of **a** the Cu (111) crystal face and **b** the Cu (111)/Cu (200) heterojunction. **c** The TDOS of Cu (111) and Cu (111)/Cu (200). **d** The optimized adsorption configurations of reaction intermediates for $C_2H_4$ and ethanol pathway on the Cu (111)/Cu (200) surface. **e** The Gibbs free energy diagrams for $CO_2$ to *$CH_2CHO$ intermediate on Cu (111) and Cu (111)/Cu (200) surface. **f** The Gibbs free energy diagrams for *$CH_2CHO$ intermediate to $C_2H_4$ and ethanol on the Cu (111)/Cu (200) surface. The *Y*-axis refers to the standard Gibbs free energy of the reaction under ambient conditions (298.15 K), pH = 0.

promote $CO_2$RR activity. It might be ascribed to the synergistic effect of the CS transition layer and the 3D structure of Cu, which facilitates rapid electron transport and produces a large number of active sites. On one hand, it was affirmed that 3D Cu–CS-GDL electrode required a significantly lower $CO_2$ reduction potential than other Cu GDEs at all the range of current densities, as shown in Supplementary Fig. S28. This indicated that the CS transition layer could reduce or eliminate the interfacial contacting resistance between the catalyst and substrate, which was favorable to decrease the applied potential of $CO_2$RR. On the other hand, electrochemical impedance spectroscopy (EIS) was also carried out to study the interfacial properties of four GDEs at an open-circuit voltage (OCV) (Supplementary Fig. S29). As shown in Supplementary Fig. S29a, it shows that the charge transfer resistance ($R_{ct}$) of 3D Cu–CS-GDL was much smaller than that of others. A reasonable interpretation of the result is that coupling 3D structure and CS can enhance electron mobility and accelerate the charge transfer rate on 3D Cu–CS-GDL interface, which is conducive to enhance the activity of $CO_2$RR. The Bode plots (Supplementary Fig. S29b) show the decrease of modulus of the impedance (log |Z|) and moving of the phase angle ($\varphi$) to the higher-frequency region of 3D Cu–CS-GDL indicates an increase of gas–liquid–solid three-phase sites, which facilities $CO_2$ diffusion and provides more opportunity for the reaction[38,39]. In addition, CS-derived adsorbents are attractive in

the $CO_2$ capture process because of the presence of amino groups in their structure[29,30]. This also can be known from the fact that when CS synergized with other components to form an integrated 3D structure, it could form abundant gas–liquid–solid three-phase interface with more exposing active sites. This phenomenon leads to the lower contact angle of 3D Cu–CS-GDL (95.1°) than that of Cu NPs-GDL (138.4°) and Cu/CS-GDL (135.8°). However, a much lower contact angle of De–Cu-GDL (70°) leads to a loss of gas diffusion ability (Supplementary Fig. S30). The above result suggests that controlling the electrode surface with an appropriate contact angle was more conducive to form an abundant gas–liquid–solid three-phase interface with more exposing active sites, which is favorable to improve the $CO_2$RR performance[38–41]. As shown in Supplementary Fig. S31a–e, the electrochemical active surface areas (ECSA) of the GDEs were estimated through the electrochemical double-layer capacitance ($C_{dl}$) measurements, and ECSA values of 3D Cu–CS-GDL, Cu NPs-GDL, De–Cu-GDL, and Cu/CS-GDL were 600, 525, 360 and 125 $cm^2_{ECSA}$, respectively[2,42]. After normalizing the current density to ECSA, 3D Cu–CS-GDL still exhibited the largest partial current densities of $C_{2+}$ alcohols at the potential of −0.87 V vs. RHE, which indicated that the 3D structure could improve the intrinsic activity for producing $C_{2+}$ alcohols in $CO_2$RR (Fig. S31f).

The results of long-term stability experiments (Fig. 3e) demonstrated that the 3D Cu−CS-GDL architecture was stable at least for 24 h at 900 mA cm$^{-2}$. After continuous CO$_2$ electrolysis, the morphology and the crystal structure of 3D Cu−CS-GDL were well preserved (Supplementary Figs. S32 and S33), suggesting that the 3D architecture could be maintained at a high current density for long-term electrolysis. We also carried out CO$_2$ electrolysis in membrane electrode assembly (MEA) (Supplementary Figs. S34 and S35)[36,43,44]. A high overall current of 1.2 A cm$^{-2}$ with C$_{2+}$ alcohols FE of 36.7% was achieved at −3.6 V cell voltage, and the production rates of EtOH and PrOH were 1.54 and 0.50 mmol h$^{-1}$ cm$^{-2}$, respectively.

In Supplementary Fig. S36, the XAFS data of 3D Cu−CS-GDL are provided at OCV, −0.4, −0.8 V vs. RHE during CO$_2$RR and after the reaction. The K-edge XANES spectra and the derivative K-edge XANES spectra indicated that 3D Cu−CS-GDL presented zero valences Cu in the whole process of CO$_2$RR. All curves in $k$ space also followed the trend of the curve of Cu foil, which also proved that Cu (0) was maintained in CO$_2$RR. In R space, the ever-present Cu−Cu bond confirmed to the above conclusion, but its corresponding radial distance was shifted with applied potential, which is caused by surface adsorption and lattice vibration in the reaction environment. Therefore, the internal bonding of 3D Cu−CS-GDL was constant.

Further seeking the reason for the improved C$_{2+}$ pathway on 3D Cu−CS-GDL electrode is very interesting. Therefore, we employed the in-situ Raman spectroscopy to study the 3D Cu−CS-GDL and Cu NPs-GDL electrodes and the key reaction intermediates in the reaction. As depicted in Supplementary Fig. S37, there was no signature peak vesting to Cu$_2$O in the range of 400−650 cm$^{-1}$, indicating that no Cu$^+$ species on the surfaces of catalysts at the reduction potentials, which is consistent with XRD, semi-in-situ XPS and XAFS results (Fig. 2g, 2h, Supplementary Figs. S6, S9, S10, and S36). It was noted that a new Raman peak located at 538 cm$^{-1}$ appeared on 3D Cu−CS-GDL electrode arose at 0.1 V vs. RHE, which was attributed to the adsorption of preliminary intermediates (such as CO$_{2ad}$ and Cu−OH band) on the active sites[45]. For Cu NPs-GDL electrode, it came out at 0 V vs. RHE, indicating that the activation of CO$_2$ molecule was easier on the 3D Cu−CS-GDL electrode than on Cu NPs-GDL electrode. At 0.1 V vs. RHE, except for 538 cm$^{-1}$, a peak can be observed at 364 cm$^{-1}$ in the Raman spectra of 3D Cu−CS-GDL electrode, corresponding to the restricted rotation of Cu−CO stretching. It was suggested that higher intensity of the Cu−CO stretching band can be assigned to a higher CO intermediate coverage and facilitate C−C coupling[46]. After −0.2 V vs. RHE, it separated into two peaks located at 305 and 380 cm$^{-1}$, respectively, and corresponded to the Cu−CO frustrated rotation and the Cu−CO stretch, then disappeared at −0.7 V vs. RHE[45,47–49]. Comparably, the two peaks of Cu NPs-GDL disappeared at a more positive potential of −0.4 V vs. RHE, indicating a stronger adsorption capacity of *CO intermediate on the 3D Cu−CS-GDL electrode. In particular, a typical peak located at 1545 cm$^{-1}$ appeared at 0.1 V vs. RHE over 3D Cu−CS-GDL electrode, which was ascribed to the important intermediate of C$_{2+}$ products[50]. However, there was no peak at 1545 cm$^{-1}$ for Cu NPs-GDL electrode, suggesting that the intermediate on the surface of 3D Cu−CS-GDL electrode could stay longer time to complete the next reaction and be favorable for the production C$_{2+}$ species. A reasonable explanation of the above results was that the 3D Cu−CS-GDL electrode could facilitate CO$_2$ diffusion, enhance the activation of CO$_2$ molecule and effectively stabilize the intermediate, leading to very high FE of C$_{2+}$ products. Otherwise, there was the C≡O stretching on Cu located about 2068 cm$^{-1}$ in Raman spectra of both 3D Cu−CS-GDL and Cu NPs-GDL, which can be deconvolved into top-bound CO and bridge-bound CO, suggesting that the pathway of generating C$_{2+}$ products was in progress[51].

In order to further verify the above conclusion, we elucidated the CO$_2$RR to C$_2$H$_4$ and ethanol pathways from a theoretical viewpoint. Taking into account the effect of the crystal interface on the electrocatalytic properties, understanding the changes arising from interfaces is critical to interpreting the enhancement of C$_2$H$_4$ or C$_{2+}$ alcohol products. We built Cu (111) and a Cu (111)/Cu (200) heterojunction as model structures of Cu NPs-GDL and 3D Cu−CS-GDL for the density functional theory (DFT) calculations (Fig. 4a and b). It was shown that electrons accumulating on the interface of the heterojunction of Cu (111)/Cu (200) are larger than that on Cu (111). The total density of states (TDOS) data (Fig. 4c) explained that the $d$ band center of Cu (111)/Cu (200) moved closer to the Fermi level, and the $d$ band of Cu (111)/Cu (200) is wider than that of pristine Cu (111). According to the $d$-band center theory, the shift of the $d$-band center for Cu (111)/Cu (200) would result in the enhanced binding strength of *CO intermediate and *OCHCHO intermediate, which in turn favors the C−C coupling process and ethanol pathway. This conclusion was also supported by the in-situ Raman analysis (Supplementary Fig. S37). The adsorption models of CO$_2$, intermediates over Cu (111) and Cu (111)/Cu (200) are shown in Supplementary Figs. S38 and S39, and the reaction pathway with them on the heterojunction are displayed in Fig. 4d, which was obtained from the Gibbs free energy diagrams (Fig. 4e). It can be seen that only 0.05 eV was required at the Cu (111)/Cu (200) surface to initiate CO$_2$RR, which is much lower than that of Cu (111) (1.22 eV). In addition, the Gibbs free energy of the entire reaction path of CO$_2$RR at the Cu (111)/Cu (200) surface was always lower than that of the reaction path on Cu (111). It demonstrated that the heterojunction between Cu (111)/Cu (200) facilitated the C$_{2+}$ pathway. It is known that the generation path of ethanol is longer than that of C$_2$H$_4$, thus it is generally easier to produce C$_2$H$_4$. However, when *CH$_2$CHO transformed to C$_2$H$_4$ or *CH$_3$CHO on the Cu (111)/Cu (200) surface, the Gibbs free energy of *CH$_3$CHO is 0.37 eV lower than that of C$_2$H$_4$ (Fig. 4f). Consequently, more *CH$_2$CHO is converted to *CH$_3$CHO, leading to more *CH$_3$CHO to stay at the heterojunction, favoring the ethanol pathway. The above results, taken together, revealed that the CS-induced growth of 3D hexagonal prismatic Cu microrods. The 3D Cu−CS-GDL structure not only facilitated rapid electron transport and mitigated CO$_2$ diffusion limitations, but also created abundant Cu (111)/Cu (200) crystal faces, which were favorable for the C$_{2+}$ alcohol pathway.

## Discussion

In summary, we found that an integrated 3D Cu−CS-GDL electrode could promote CO$_2$ electrocatalysis to C$_{2+}$ alcohols at a large current density. C$_{2+}$ FE of 88.2 % with a current density as high as 900 mA cm$^{-2}$ could be reached at the potential of −0.87 V vs. RHE, of which the C$_{2+}$ alcohols selectivity was 51.4% with a partial current density of 462.6 mA cm$^{-2}$. The outstanding electrocatalytic performance of the catalyst could be ascribed to using CS as "transition layer", which induced the growth of the 3D hexagonal prismatic Cu microrod film with abundant Cu (111)/Cu (200) crystal faces. The novel structure of the electrode enhanced both mass transformations, but also for the C$_{2+}$ alcohol pathway. We believe that the use of a suitable transition layer in the GDEs to tune the architecture is applicable to the design of other efficient electrodes for CO$_2$RR.

## Methods
### Materials
CuCl$_2$·2H$_2$O (A.R. grade), chitosan (low viscosity, 100−200 mPa s), polyvinylpyrrolidone (PVP, K30), acetone (A.R. grade), hydrogen chloride (HCl, 37%), Ni foam, Al foil (2 mm) were provided by Sinopharm Chemical Reagent Co., Ltd, China. Potassium bicarbonate (KHCO$_3$, >99.7%), potassium hydroxide (KOH, A.R. grade, 85%) were purchased from Acros. Sodium borohydride (NaBH$_4$, 98%), polytetrafluoroethylene-hydrophobized carbon paper (PTFE-CP, Toray, YLS-30T GDL), and Nafion D-521 dispersion (5% w/w in water and 1-propanol, ≥0.92 meg/g exchange capacity) were purchased from Alfa Aesar China Co., Ltd. Anion-exchange membrane (3PK-130-100 × 100)

was purchased from Gaoss Union Company. $CO_2$ (99.999%) was provided by Beijing Analytical Instrument Company.

## The preparation of 3D Cu−CS-GDL electrode

Briefly, dispersing chitosan (0.200 g) evenly in 20 mL of distilled water. Then, 10 mL of 12 mM $CuCl_2$ solution was added to the above suspension with stirring at 40 °C. After 6 h, Cu−CS complexes were filtered and washed with water several times and then dried under vacuum at 60 °C overnight. The Cu (II) contents of Cu−CS were determined using inductively coupled plasma optical emission spectroscopy (ICP-OES) analyses, which was 3.51 wt%.

15 mg of dried Cu−CS was dispersed in 1 mL of acetone and then evenly dripped onto six pieces of $1 \times 2\ cm^2$ PTFE-CP. Subsequently, 100 mA current was maintained for 1000 s at constant current mode in $CO_2$-saturated 1 M $KHCO_3$ electrolyte. The surface color of CP changed from blue to dark red, resulting in pre-Cu−CS-1. After that, pre-Cu−CS-2 was obtained by electrodeposition in 0.15 M $CuCl_2$/0.5 M $HCl$/$H_2O$ electrolyte for 600 s at an applied potential of −1.3 V vs. Ag/AgCl. Rinsing pre-Cu−CS-2/CP with deionized water and acetone and drying it by blowing. Finally, the 3D Cu−CS-GDL electrode was generated in situ after the first minute of $CO_2$RR in a flow cell.

## The preparation of De−Cu·GDL electrode

PTFE-CP directly in 0.15 M $CuCl_2$/0.5 M $HCl$/$H_2O$ electrolyte for electrodeposition for 600 s at an applied potential of −1.3 V vs. Ag/AgCl. The sample was washed with deionized water and acetone and then dried by blowing. Then, De−Cu was generated in situ after the first minute of $CO_2$RR in a flow cell.

## The preparation of Cu/CS composite-GDL electrode

Dispersing the chitosan (0.200 g) evenly in 20 mL of distilled water. Then, 10 mL 12 mM $CuCl_2$ solution was added to this suspension with stirring at 40 °C. After 6 h, 1 mL of freshly prepared $NaBH_4$ (22.7 mg) aqueous was dropped slowly into the above suspension for 10 min, followed by 2 h of stirring at room temperature. The resulting purple mixture was centrifuged and washed several times with deionized water and acetone. Cu/CS was dried in a vacuum oven at 60 °C overnight. 15 mg Cu/CS powder prepared above was suspended in 1 mL acetone with 20 µL Nafion D-521 dispersion (5 wt%) via ultrasound. Then, the suspension was evenly drop-coated onto six pieces of $1 \times 2\ cm^2$ PTFE-CP, uniformly spreading, obtaining Cu/CS-GDL electrode.

## The preparation of Cu NPs-GDL electrode

Cu NPs were obtained by hydrothermal method in reference to the previous literature[31]. Firstly, 0.5 mmol $CuCl_2 \cdot 2H_2O$ and 150 mg PVP were added to 60 mL deionized water and 20 mL absolute ethanol to prepare solution A, which was stirred in a 250 mL three-mouth flask under $N_2$ atmosphere at 40 °C. 1 mmol $NaBH_4$, 1 mmol KOH, and 150 mg PVP were added to 60 mL deionized water to prepare solution B. Subsequently, solution B was dropped slowly into solution A, followed by two hours magnetically stirring at 40 °C. The black precipitate was centrifuged and washed several times with deionized water and acetone. Cu NPs were dried in a vacuum oven at 60 °C overnight. 15 mg Cu NPs powder prepared above was suspended in 1 mL acetone with 20 µL Nafion D-521 dispersion (5 wt%) via ultrasound. Then, the suspension was evenly drop-coated onto six pieces of $1 \times 2\ cm^2$ PTFE-CP, uniformly spreading, obtaining Cu NPs-GDL electrode.

## Characterizations

The microstructures of the as-synthesized materials were characterized by SEM (HITACHI S-4800) and TEM (JEOL-2100F) equipped with an energy dispersive spectrometer (EDS). XRD analysis was performed on a Rigaku D/max-2500 diffractometer with Cu Kα radiation ($\lambda = 1.5418$ Å) at 40 kV and 200 mA. Quasi-in-situ XPS study was carried out on a Thermo Scientific ESCALab 250Xi using 200 W Al-Kα radiation. The elemental

contents of the catalysts were detected using inductively coupled plasma optical emission spectroscopy (ICP-OES, Vista-MPX). Fourier transform infrared (FT-IR) spectra were recorded using a Bruker VERTEX 70 V spectrometer and the samples were prepared by the KBr pellet method. In situ Raman spectroscopy (Horiba Labram HR Evolution Raman System) was conducted in a modified flow cell using a 785-nm excitation laser and signals were recorded using a 20 s integration and by averaging two scans. The signals were recorded at different applied potentials, and a 5 min electrolysis was conducted to gain the steady state before the collection of Raman spectra with constantly flowed gaseous $CO_2$. X-ray adsorption spectroscopy (XAS) experiments were carried out at Beamline 1W1B at the Beijing Synchrotron Radiation Facility. The static contact angles were measured using an OCA20 apparatus (Data-Physics, Germany) and 10 µL water droplets.

## Quasi in situ X-ray photoelectron spectra (XPS) measurement

The Quasi in situ XPS were measured on a Thermo Scientific ESCALab 250Xi using 200 W Al-Kα radiation. For investigating the evolution of Cu and N species in the reaction process, catalysts were electrolyzed at different times in 1 M KOH. After that, the samples were washed with acetone immediately and put into the glove box. They were cut into $4 \times 4\ mm^2$ and glued on a self-developed quasi-in-situ XPS sample stage with a double-sided adhesive, followed by evacuation to prevent the samples to be oxidized in the air. Finally, the stage was transferred to the XPS chamber for measurement.

## Electrocatalysis experiments

All the electrochemical experiments were conducted on the electrochemical workstation (CHI 660E, Shanghai CH Instruments Co., China) equipped with a high current amplifier CHI 680c. The electrocatalysis experiments were carried out in a separated flow cell with three chambers. As shown in Supplementary Fig. S19, the cathode chamber included the as-synthesized electrodes as working electrodes (WE) and Hg/HgO electrode (with 1 M KOH used as the filling solution) as reference electrode (RE), in which the gas chamber was separated from the cathode chamber by WE. $CO_2$ was then diffused to the catholyte through a gas diffusion layer and the electrolysis occurs at the gas−liquid−solid three-phase interface. A piece of Ni foam was used as a counter electrode (CE) at the anode chamber. The cathode and anode chambers were separated by an anion-exchange membrane. All the electrodes and anion-exchange membranes were fixed and sealed by a silicone pad, and their effective area for the electrolytic reaction was $0.5 \times 2\ cm^2$. In the experiments, all potentials were converted to the RHE reference scale using the following relation and compensated with the solution resistance.

$$E_{RHE} = E_{Hg/HgO} + 0.098 + 0.059 \times pH \tag{1}$$

1 M KOH was used as the electrolyte, which was circulated through the cathodic and anodic chambers using peristaltic pumps at 10 mL min$^{-1}$. The flow rate of $CO_2$ gas through the gas chamber was controlled to be 50 sccm using a digital gas flow controller. The linear sweep voltammetry (LSV) curves were obtained at a sweep rate of 20 mV s$^{-1}$.

## Electrochemical active surface area (ECSA) measurements

The ECSA was calculated as follows using the reported method[2,42]:

$$ECSA = R_f S \tag{2}$$

in which $S$ stands for the real surface area of the smooth metal electrode, which was generally equal to the geometric area of the glassy carbon electrode (in this work, $S = 1\ cm^{-2}$). The roughness factor $R_f$ was estimated from the ratio of the double-layer capacitance ($C_{dl}$) for WE and the corresponding smooth metal electrode (assuming that the

average double-layer capacitance of a smooth metal surface is 20 μF cm$^{-2}$), and it is $R_f = C_{dl}/20\ \mu F\ cm^{-2}$. Cyclic voltammogram (CV) measurements of the catalysts were conducted with various scan rates to obtain $C_{dl}$. The $C_{dl}$ was estimated by plotting the $\Delta j(j_a - j_c)$ at middle potential versus Hg/HgO against the scan rates, in which $j_a$ and $j_c$ are the anodic and cathodic current densities, respectively. The linear slope was equivalent to twice of the $C_{dl}$ (Fig. S31a−e).

## CO₂ electrolysis in membrane electrode assembly (MEA)

The MEA electrolysis was conducted with an as-prepared 3D Cu-CS-GDL electrode ($1 \times 1\ cm^2$) as the cathode and $IrO_2$/CP as the anode ($1 \times 1\ cm^2$). 0.1 M KOH aqueous solution was used as the anolyte and circulated using a pump at a rate of 40 mL min$^{-1}$. On the cathode side, $CO_2$ gas (30 sccm) was continuously humidified with DI water and fed into the cathode chamber. The performance of the catalysts in an MEA system was evaluated by applying different cell voltages. Due to the liquid product crossover, the FEs of liquid products were calculated using the total amount of the products at anodes and cathodes. In the cathode, liquid products were collected using a cold trap.

## Product analysis

After the electrolysis reaction, the gaseous products were collected using a gas bag and then analyzed by an Agilent 4890 gas chromatograph (GC) equipped with a TCD detector using helium as the internal standard. The cathode liquid products were analyzed by $^1$H NMR using a Bruker Avance III 400 HD spectrometer. The Faradaic efficiency (FE) of the cathode products was calculated using GC and NMR data.

After the quantification, the FE of each product was calculated as follows:

$$FE(\%) = \frac{n \times F \times \text{moles of product}}{Q} \times 100\% \qquad (3)$$

($Q$: the amount of charge passed through the working electrode; $F$: The Faraday constant (96,485 C mol$^1$); $n$: number of moles of electrons to participate in the Faradaic reaction, for the $H_2$, CO, $C_2H_4$, HCOOH, $CH_3COOH$, EtOH and PrOH, the $n$ is 2, 2, 12, 2, 8, 12 and 18, respectively.

## Computational method

The density functional theory (DFT) calculations were performed by employing the Vienna ab-initio simulation package (VASP) with spin polarization. The interactions between the electrons and ions were described using the projector-augmented-wave (PAW) method, while the Perdew–Burke–Ernzehof (PBE) functional within the generalized gradient approximation (GGA) was utilized to treat the electronic exchange-correlation energies. The electron–ion interactions were described by the projector-augmented wave method, and the cut-off energy for the plane-wave basis set was 500 eV. $4 \times 4 \times 1$ $k$-points using the Monkhorst-Pack scheme grid for geometry optimization, the self-consistent calculations were used to sample the Brillouin zone. To avoid interaction between two periodic units, a vacuum space exceeding 10 Å was employed. All atoms could relax until the electronic self-consistence and the ionic relaxation reaching the convergence criteria of $10^{-5}$ eV and 0.02 eV/Å.

Gibbs free energy change ($\Delta G$) is defined as

$$\Delta G = \Delta E + \Delta E_{ZPE} + \Delta \int C_P dT - T\Delta S \qquad (4)$$

where $\Delta E$, $\Delta E_{ZPE}$, $\Delta \int C_P dT$ and $\Delta S$ are the total energy difference, the zero-point energy difference, the difference in enthalpic correction and the entropy change between the products and reactants obtained from DFT calculations, respectively. The zero-point energies (ZPE) and total entropies of the gas phase were computed from the vibrational frequencies, and the vibrational frequencies of adsorbed species were also computed to obtain the ZPE contribution in the free energy expression. Only vibrational modes of the adsorbates were computed explicitly, while the catalyst sheet was fixed (assuming that vibration contribution to the free energy from the substrate is negligible). $T$ is the temperature (298.15 K).

## Data availability

The authors make a statement that the data presented by this article are available from the corresponding author on reasonable requests. Source data are provided with this paper.

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

## Acknowledgements

The work was supported by the National Natural Science Foundation of China (22022307, 22279146, 22102192, 22033009, 21890761, 21733011 and 22121002), CAS Project for Young Scientists in Basic Research (Grant No. YSBR-050), the National Key Research and Development Program of China (2020YFA0710203), Beijing Natural Science Foundation (J210020), S&T Program of Hebei (B2021208074), Chinese Academy of Sciences (QYZDY-SSW-SLH013) and Photon Science Center for Carbon Neutrality.

## Author contributions

J.H.B., Q.G.Z., and B.X.H. proposed the project, designed the experiments, and wrote the manuscript. J.H.B. performed the whole experiment. P.S.L., J.Y.L., S.Q.J., Y.W., and Z.M.L. performed the analysis of experimental data. Q.G.Z. and B.X.H. co-supervised the whole project. All authors discussed the results and commented on the manuscript.

## Competing interests

The authors declare no competing interests.
