## [Peer Review File · Nature Communications]

REVIEWER COMMENTS

Reviewer #1 (Remarks to the Author):

Using chitosan as an intermediate layer in CO₂RR GDEs, the authors achieved high C₂+ FEs, particularly for C₂+ alcohols. The manuscript is timely and presents important CO₂RR performance with numerous experimental catalyst materials characterisation results and theoretical studies to elucidate the mechanism of CO₂ conversion. There is a need to clarify the use and significance of current densities, in particular, ECSA-normalised current densities should be used for what is to be regarded as the intrinsic activity of the electrocatalysts (<https://pubs.acs.org/doi/10.1021/acscatal.8b01340>), as detailed below.

Line 122: The statement "which was not only stabilized the hydrophobic surface of the GDL" is not clear. Do the authors mean "which was not only stabilized by the hydrophobic surface of the GDL" or "which was not only stabilizing the hydrophobic surface of the GDL"? Please clarify.

Figure 2g and all other XRD patterns presented in the study: What is the main peak at about 24 2Theta due to? And the other two smaller peaks around it?

Line 145: Is Cu/CS composite made of "Cu NPs + CS" or "Cu ions + CS"?

Line 150: The authors state "Figure S12 illustrated that Cu NPs were uniformly distributed in the CS network structure", but where can Cu NPs be seen in Figure S12? There may be some NPs visible in image "e" but it is not apparent.

Line 172: "1 M KOH aqueous solution" in the air or saturated with CO₂? Please specify.

Line 211: Where are the values of ECSA? Figure S26 reports the values of electric double-layer capacitance for different GDEs but not of electrochemical surface area.

Line 214: The statement "It was obvious that 3D Cu-CS-GDL electrode had the largest ECSA, indicating that the 3D Cu structure was responsible for promoting high CO₂RR activity via generating more active sites" may need to be revised based on the following considerations. The ECSA-

normalised current density represents the CO₂RR activity per unit of electroactive area (also referred to as "intrinsic activity"). A larger electrochemical surface area corresponds to a lower current density, hence lower CO₂RR activity per unit area. For example, assuming that the values in Figure S19 are geometric current densities (please specify), the relative values of the LSV curves change when ESCA-normalised currents are used. In particular in Figure S19, the geometric current density of Cu-CS-GDL and Cu/CS-GDL are about -0.6 and -0.5 A/cm², respectively, at -0.6V. However, the ESCA of Cu-CS-GDL should be about 5 times larger than that of Cu/CS-GDL (based on the ratio of Cdl values 12.0:2.5, from Figure S26e), it follows that the ESCA-normalised current density of Cu-CS-GDL is significantly lower than that of Cu/CS-GDL, meaning that the intrinsic activity of Cu-CS-GDL is less than that of Cu/cs-GDL. This is a critical point that needs to be clarified in the manuscript, for all electrocatalysts using their corresponding ECSA-normalised current densities.

In all cases, please specify which current densities are geometric or ECSA normalised.

In all cases, please use either "Cu-CS-GDL" or "3D Cu-CS-GDL" but not both otherwise there is uncertainty regarding being either the same GDE or two different ones.

Supporting information

In addition to the experimental details given in the manuscript, please provide a detailed description (possibly with photos) of the flow cell reactor used for CO₂ electrolysis. In particular, provide the dimensions of the CO₂RR GDE, and which ion transport membrane was used.

Figure S8 & S12: What are the rods seen in "e"?

Reviewer #2 (Remarks to the Author):

This study reports a 3D Cu-chitosan (CS)-GDL electrode for CO₂ reduction. The authors claimed that the CS acts as a transition layer between catalyst and the GDL which can facilitate rapid electron transport and mitigate mass diffusion limitations. This design resulted in a C₂⁺ Faradaic efficiency (FE) of 88.2% at 900 mA/cm² with a C₂⁺ alcohols selectivity of 51.4% and a notable partial current density of 462.6 mA/cm². I see value in the work. However, the following must be addressed for the work to be suitable for Nature Communications.

Major comments:

1- Adding a polymer layer on top of GDL can significantly decrease the electrical conductivity and impeding the electron transfer between the GDL and catalyst. More experiment/characterization and explanation are needed to support the authors' claim: "the CS acts as a transition layer between catalyst and the GDL which can facilitate rapid electrons transport and mitigates mass diffusion limitations".

2- It is not clear the extent to which this layer differs – in make-up and function – from the many polymer, ionomer, binder etc. layers that are commonly applied in this field.

3- A high EtOH selectivity at high current densities was achieved in a flow cell. Achieving record performance in membrane electrode assembly (MEA) would support the claim that this is a broadly applicable approach and one that advances the field toward practical application.

4- This study was performed in an alkaline flow cell which leads to a severe carbonate formation, and the many associated issues documented well in this journal by the perspective by Kanan. How can the authors address the Kanan's concerns?

5- Most of the commercially available GDLs are hydrophobic. The reported contact angle in this study (95.1) is less than some previous reports. More explanation is needed to support why Cu-chitosan (CS)-GDL mitigates mass diffusion limitations and improves the CO₂ reduction performance at higher current densities.

6- Cu NPs-GDL electrode shows a good performance at higher current densities (600-1000 mA/cm²). Is this related to the synthesis approach? A comparison between commercial CuNP and the Cu NPs-GDL is needed.

7- A deeper mechanistic investigation is needed to support the high EtOH FE. For example, Raman analysis can be compared to these works to further investigate the reaction pathways: J. Am. Chem. Soc. 2019, 141, 21, 8584, Nat Catal 3, 75–82 (2020), J. Mater. Chem. A, 2022, 10, 20059-20070.

8- XAS analysis shows some interesting results. I do recommend revisiting the results and combining them with the mechanistic study in the main text.

9- XRD needs to be revised, I suggest removing graphitic peaks and focus on the Cu peaks. If the interface of Cu (200) and Cu (111) is very important in EtOH pathway and production, this needs to be experimentally shown.

10- The results in this paper are impressive but do not exceed state-of-the-art performance to my understanding. E.g: Small Methods 2022, 6 (2), 2101334, Joule 2021, 5 (2), 429.

Minor

1- The authors claimed that "Unfortunately, construction of 3D nanostructured catalysts on porous gas diffusion layer (GDL) is very difficult". Spraying a mixture of a catalyst and ionomer (which is very common) leads to a 3D structure on GDL. Therefore, making a 3D structure is not challenging.

2- In some case, the total FE is higher than 100%, indicating an error.

Responses to the comments and revisions made

Reviewer #1 (Remarks to the Author):

Comments to the Author

Using chitosan as an intermediate layer in CO₂RR GDEs, the authors achieved high C₂₊ FEs, particularly for C₂₊ alcohols. The manuscript is timely and presents important CO₂RR performance with numerous experimental catalyst materials characterization results and theoretical studies to elucidate the mechanism of CO₂ conversion. There is a need to clarify the use and significance of current densities, in particular, ECSA-normalized current densities should be used for what is to be regarded as the intrinsic activity of the electrocatalysts (<https://pubs.acs.org/doi/10.1021/acscatal.8b01340>), as detailed below.

Response: We thank the reviewer very much for positive and valuable comment. We have addressed all the concerns, including those about ECSA, as can be known from the answers to the following detailed comments.

Comment 1: Line 122: The statement "which was not only stabilized the hydrophobic surface of the GDL" is not clear. Do the authors mean "which was not only stabilized by the hydrophobic surface of the GDL" or "which was not only stabilizing the hydrophobic surface of the GDL"? Please clarify.

Response 1: We thank the reviewer for the comment. In the revised manuscript, we have clarified it to "which not only stabilized the hydrophobic surface of the GDL". Please see **Page 6** in the revised manuscript.

Comment 2: Figure 2g and all other XRD patterns presented in the study: What is the main peak at about 24 2Theta due to? And the other two smaller peaks around it?

Response 2: We thank the reviewer for the comment. As there is a layer of graphite in the hydrophobic carbon paper, the main peak mentioned by the reviewer was due to the (002) lattice plane of graphite, which was located at 26.1° in the XRD patterns. The other two smaller peaks around it were also attributed to graphite. To avoid

confusion, in the revised manuscript, we only give the pattern in the useful angle range from 30° to 60° , which shows Cu peaks clearly. Please see **Figure 2g, S9, S13, S17 and S33** in the revised manuscript.

Comment 3: Line 145: Is Cu/CS composite made of "Cu NPs + CS" or "Cu ions + CS"?

Response 3: We thank the reviewer for the comment. The Cu/CS composite was made of "Cu NPs + CS", as can be known from the characterizations in the manuscript (**Figure S11-13**). We have also emphasized this by **"The Cu/CS composite was made of Cu NPs and CS."** Please see **Page 7** in the revised manuscript.

Comment 4: Line 150: The authors state "Figure S12 illustrated that Cu NPs were uniformly distributed in the CS network structure", but where can Cu NPs be seen in Figure S12? There may be some NPs visible in image "e" but it is not apparent.

Response 4: We thank the reviewer for the comment. After reading the comment, we have added the TEM with high-magnification and the particle size distribution of Cu NPs in catalyst. As shown in **Figure S12f and g**, The TEM images indicate that the average Cu NPs size was around 5 nm, and Cu NPs were uniformly distributed in the CS network structure. Please see **Page 11** in the Supplementary information of the revised manuscript.

In the revised manuscript, we have also emphasized this by **"Supplementary Figure S12f and g further illustrated that Cu NPs with average particle size around 5 nm were uniformly distributed in the CS network structure."** Please see **Page 7** in the revised manuscript.

Comment 5: Line 172: "1 M KOH aqueous solution" in the air or saturated with CO₂? Please specify.

Response 5: We thank the reviewer for the comment. As the electrolysis was carried out in a separated flow cell with three chambers, 1 M KOH aqueous solution was in the air and CO₂ was then diffused to the catholyte through a gas diffusion layer.

According to the comment, in the revised manuscript, we have made note in the section of “**Electrocatalysis experiments**” and discussed this by “The electrocatalysis experiments were carried out in a separated flow cell with three chambers. As shown in **Supplementary Figure S19**, the cathode chamber included the as-synthesized electrodes as working electrode (WE) and Hg/HgO electrode (with 1 M KOH used as the filling solution) as reference electrode (RE), in which the gas chamber was separated from the cathode chamber by WE. CO₂ was then diffused to the catholyte through a gas diffusion layer and the electrolysis occurs at the gas-liquid-solid three-phase interface. A piece of Ni foam was used as counter electrode (CE) at the anode chamber. The cathode and anode chambers were separated by an anion-exchange membrane. All the electrodes and anion-exchange membrane were fixed and sealed by silicone pad, and their effective area for the electrolytic reaction was $0.5 \times 2 \text{ cm}^2$.” Please see **Page 19** in the “**Electrocatalysis experiments**” section of the revised manuscript. Accordingly, the diagram of the flow cell we used was also added, which was shown in **Figure S19**.

Comment 6: Line 211: Where are the values of ECSA? Figure S26 reports the values of electric double-layer capacitance for different GDEs but not of electrochemical surface area.

Response 6: We thank the reviewer for the comment. As suggested by the referee, in the revised manuscript, we have calculated the ECSA, and the specific calculation method is supplemented in the “**Electrochemical active surface area (ECSA) measurements**” section. Please see **Page 20** in the revised manuscript.

In the revised manuscript, we have also discussed this by “As shown in **Supplementary Figure S31a-e**, the electrochemical active surface areas (ECSA) of the GDEs were estimated through the electrochemical double-layer capacitance (C_{dl}) measurements, and ECSA values of 3D Cu-CS-GDL, Cu NPs-GDL, De-Cu-GDL and Cu/CS-GDL were 600, 525, 360 and 125 $\text{cm}^2_{\text{ECSA}}$, respectively.^{2, 42}”. Please see **Page 11** in the revised manuscript.

Comment 7: Line 214: The statement "It was obvious that 3D Cu-CS-GDL electrode had the largest ECSA, indicating that the 3D Cu structure was responsible for promoting high CO₂RR activity via generating more active sites" may need to be revised based on the following considerations. The ECSA-normalised current density represents the CO₂RR activity per unit of electroactive area (also referred to as "intrinsic activity"). A larger electrochemical surface area corresponds to a lower current density, hence lower CO₂RR activity per unit area. For example, assuming that the values in Figure S19 are geometric current densities (please specify), the relative values of the LSV curves change when ESCA-normalised currents are used. In particular in Figure S19, the geometric current density of Cu-CS-GDL and Cu/CS-GDL are about -0.6 and -0.5 A/cm², respectively, at -0.6 V. However, the ESCA of Cu-CS-GDL should be about 5 times larger than that of Cu/CS-GDL (based on the ratio of Cdl values 12.0:2.5, from Figure S26e), it follows that the ESCA-normalised current density of Cu-CS-GDL is significantly lower than that of Cu/CS-GDL, meaning that the intrinsic activity of Cu-CS-GDL is less than that of Cu/cs-GDL. This is a critical point that needs to be clarified in the manuscript, for all electrocatalysts using their corresponding ECSA-normalised current densities.

In all cases, please specify which current densities are geometric or ECSA normalised.

Response 7: We thank the reviewer for the very instructive comment. In the revised manuscript, we have measured the ECSA and the current densities were also ECSA normalized. The results were provided in Supplementary **Figure S31f**. After normalizing the current density to ECSA, 3D Cu-CS-GDL still exhibited the largest partial current densities of C₂₊ alcohols at the potential of -0.87 V vs RHE, which indicates that the 3D structure could improve the intrinsic activity of the catalyst.

In the revised manuscript, we have revised the discussion by “**After normalizing the current density to ECSA, 3D Cu-CS-GDL still exhibited the largest partial current densities of C₂₊ alcohols at the potential of -0.87 V vs RHE, which indicates that the 3D structure could improve the intrinsic activity for producing C₂₊ alcohols in CO₂RR (Figure S31f).**”. Please see **Page 11** in the revised manuscript.

In other cases, we have used current densities with geometric normalized, in order

to compare it with other catalysts in the literature. In the revised manuscript, we have emphasized this by “the C₂₊ Faradaic efficiency (FE) could reach 88.2% with a current density (geometrically normalized) as high as 900 mA • cm⁻²” and “Comparison of FE of C₂₊ products and current density (*j*, geometric normalized) over 3D Cu-CS-GDL architecture with some typical Cu-based catalysts in CO₂RR”. Please see **Page 1**, **Page 3** and **Table S1** in the revised manuscript.

Comment 8: In all cases, please use either "Cu-CS-GDL" or "3D Cu-CS-GDL" but not both otherwise there is uncertainty regarding being either the same GDE or two different ones.

Response 8: We thank the reviewer for the comment. According to the comment, we have replaced "Cu-CS-GDL" with "3D Cu-CS-GDL" in the revised manuscript.

Supporting information

Comment 9: In addition to the experimental details given in the manuscript, please provide a detailed description (possibly with photos) of the flow cell reactor used for CO₂ electrolysis. In particular, provide the dimensions of the CO₂RR GDE, and which ion transport membrane was used.

Response 9: We thank the reviewer for the comment. According to the comment, the diagram of the flow cell we used was added, which was shown in **Figure S19**.

In the revised manuscript, we have also added the description by “The electrocatalysis experiments were carried out in a separated flow cell with three chambers. As shown in **Supplementary Figure S19**, the cathode chamber included the as-synthesized electrodes as working electrode (WE) and Hg/HgO electrode (with 1 M KOH used as the filling solution) as reference electrode (RE), in which the gas chamber was separated from the cathode chamber by WE. CO₂ was then diffused to the catholyte through a gas diffusion layer and the electrolysis occurs at the gas-liquid-solid three-phase interface. A piece of Ni foam was used as counter electrode (CE) at the anode chamber. The cathode and anode chambers were separated by an

anion-exchange membrane. All the electrodes and anion-exchange membrane were fixed and sealed by silicone pad, and their effective area for the electrolytic reaction was $0.5 \times 2 \text{ cm}^2$.". Please see **Page 19** in the “**Electrocatalysis experiments**” section of the revised manuscript.

Comment 10: Figure S8 & S12: What are the rods seen in "e"?

Response 10: We thank the reviewer for the comment. The rods seen in **FigureS8e** and **Figure S12e** are graphite (or carbon fiber) in the hydrophobic carbon paper, which is represented by gray rods in the schematic illustration of **Figure 1**.

Reviewer #2 (Remarks to the Author):

Comments to the Author

This study reports a 3D Cu-chitosan (CS)-GDL electrode for CO₂ reduction. The authors claimed that the CS acts as a transition layer between catalyst and the GDL which can facilitate rapid electron transport and mitigate mass diffusion limitations. This design resulted in a C₂⁺ Faradaic efficiency (FE) of 88.2% at 900 mA/cm² with a C₂⁺ alcohols selectivity of 51.4% and a notable partial current density of 462.6 mA/cm². I see value in the work. However, the following must be addressed for the work to be suitable for Nature Communications.

Response: We thank the reviewer for the very instructive comments. We have addressed all the concerns, as can be known from the answers to the following detailed comments.

Major comments:

Comment 1: Adding a polymer layer on top of GDL can significantly decrease the electrical conductivity and impeding the electron transfer between the GDL and catalyst. More experiment/characterization and explanation are needed to support the authors' claim: "the CS acts as a transition layer between catalyst and the GDL which can facilitate rapid electrons transport and mitigates mass diffusion limitations".

Response 1: We thank the reviewer for the comment. After reading the comment, we have made the following modifications.

(i) To elucidate this point, we use electrochemical impedance spectroscopy (EIS) to study the interfacial properties of the electrodes (**Figure S29**). On one hand, **Figure S29a** shows the Nyquist plot of various electrodes. It indicates that the charge transfer resistance (R_{ct}) of 3D Cu-CS-GDL was much smaller than that of others. A reasonable interpretation of the result is that coupling 3D structure and CS can enhance electron mobility and accelerate the charge transfer rate on 3D Cu-CS-GDL interface, which is conducive to enhance the activity of CO₂RR. On the other hand, the Bode plots (**Figure S29b**) show the decrease of modulus of the impedance ($\log |Z|$) and moving of the phase angle (ϕ) to higher-frequency region of 3D Cu-CS-GDL, indicating

increase of gas-liquid-solid three-phase sites, which facilitates CO₂ diffusion and provides more opportunity for the reaction (*Nat Catal* **18**, 1222-1227 (2019); *Nat Catal* **20**, 1000-1006 (2021)).

(ii) The enhancing diffusion may also originate from the unique properties of CS, as the good CO₂ adsorption capacity of CS has been reported (*Ind Eng Chem Res* **61**, 10522-10530 (2022); *ACS Sustainable Chem Eng* **5**, 10379-10386 (2017); *Polymer J* **34**, 144-148 (2002)). It has been found that CS-derived adsorbents are attractive in the CO₂ capture process because of the presence of amino groups in their structure. In addition, some researchers have made innovative applications in environmental, medical and other fields by using the capture CO₂ ability of CS. Therefore, when CS synergized with other components to form integrated structure, the electron transport and mitigates mass diffusion could be improved.

In the revised manuscript, we have added the discussion by “On the other hand, the electrochemical impedance spectroscopy (EIS) was also carried out to study the interfacial properties of four GDEs at an open-circuit voltage (OCV) (**Supplementary Figure S29**). As shown in **Supplementary Figure S29a**, the charge transfer resistance (R_{ct}) of 3D Cu-CS-GDL was much smaller than that of others. A reasonable interpretation of the result is that coupling 3D structure and CS can enhance electron mobility and accelerate the charge transfer rate on 3D Cu-CS-GDL interface, which is conducive to enhance the activity of CO₂RR. The Bode plots (**Supplementary Figure S29b**) show the decrease of modulus of the impedance ($\log |Z|$) and moving of the phase angle (φ) to higher-frequency region of 3D Cu-CS-GDL, indicates increase of gas-liquid-solid three-phase sites, which facilitates CO₂ diffusion and provides more opportunity for the reaction.^{38, 39} In addition, CS-derived adsorbents are attractive in the CO₂ capture process because of the presence of amino groups in their structure.^{29, 30}”. Please see **Page 10-11** in the revised manuscript.

Comment 2: It is not clear the extent to which this layer differs – in make-up and function – from the many polymers, ionomer, binder etc. layers that are commonly applied in this field.

Response 2: We thank the reviewer for the comment. After reading the comment, we have discussed the difference between CS and commonly used polymers/binders, such as polyaniline (PANI), polypyrrole (PPy) and Nafion D-521. Particularly, PANI and PPy can be considered as conductive polymers commonly investigated in electrochemical experiments, because of largely improving the electron-transfer ability (*Angew Chem Int Ed* **58**, 15834-15840 (2019); *ACS Appl Mater Interfaces* **13**, 54959-54966 (2021); *ACS Catal* **10**, 4103-4111 (2020); *Nat Catal* **19**, 266-276 (2020); *Energy Environ Sci* **9**, 1687-1695 (2016); *Angew Chem Int Ed* **60**, 10977-10982 (2021)). The N-containing groups could stabilize metal particles and reaction intermediates, enriching electrons and reactants at metal active sites. Nafion D-521 dispersion is also a kind of polymeric binders (*Science* **365**, 367-369 (2019) *Chemistry* **28**, e202200242 (2022); *Nat Catal* **4**, 20-27 (2020); *Nature* **575**, 639-642 (2019)). However, When using these polymers/binders to prepare electrochemical devices, the most direct way to prepare electrodes is mixed powder-type electrocatalysts with polymeric binders and drop-coating on the substrate electrode. The limited number of active sites, and the inert base with low mass and low electron transfer remains the major disadvantage of most metal polymer catalysts. Eventhough this method is simple, easy to operate, and the catalyst can be more firmly fixed on the substrate. However, the additive binders would inevitably decrease the CO₂RR performance and considerably increase the overpotential, which are due to the obstruction of gas transport, insufficient exposure of active sites, and detachment of catalyst from electrode surface by binder degradation in the reaction (*Angew Chem Int Ed* **58**, 4031-4035 (2019); *Chemistry* **28**, e202200242 (2022); *Nature* **575**, 639-642 (2019)).

In this work, we use CS for several reasons. First, CS is different from the source of polymers and binders mentioned above. It is a sustainable and environmental-friendly biomass resource. It is obtained from the carapaces of shrimp and crabs, containing a carbon skeleton with amino functional groups, which has the advantages of low cost, non-toxic, renewable, degradable and abundant reserves. The hydroxyl group and amino group in CS structure make it have strong affinity,

especially has good chelation ability for transition metals and rare earth metals, coordinating with metal ions to form complexes, this property also provides a basis for dispersing metal active sites (*Appl Catal B Environ* **255**, 117740 (2019)). Second, the affinity of CS to Cu is higher than that to other metals (*Int J Biol Macromol* **75**, 346-353 (2015); *J Agric Food Chem* **64**, 6148-6155 (2016); *Catal Lett* **149**, 2089-2097 (2019)). Third, CS has been proved to have the ability of structure guidance and good adsorption of CO₂ (*Ind Eng Chem Res* **61**, 10522-10530 (2022); *Biomacromolecules* **5**, 2340-2346 (2004)). Consequently, we think it is interesting to develop a new paradigm for CS application and propose a new route to prepare 3D Cu-CS-GDL electrode for CO₂ reduction, and we believe that novel use of CS in the GDEs to tune the architecture is applicable to design of other efficient electrodes for CO₂RR.

In the revised manuscript, we have added the discussion by “For catalyst layer, the most straight forward way is coating of powder-type electrocatalysts onto a gas diffusion layer (GDL) using commonly used polymers/binders, such as polyaniline (PANI), polypyrrole (PPy) and Nafion D-521.¹²⁻¹⁴ However, the additive binders would inevitably decrease the CO₂RR performance and considerably increase the overpotential, which are due to the obstruction of gas transport, insufficient exposure of active sites, and detachment of catalyst from electrode surface by binder degradation in the reaction.¹⁵⁻¹⁷” and “Chitosan (CS), an abundant amino polysaccharide, obtained from the carapaces of shrimp and crabs, containing a carbon skeleton with amino functional groups.²⁵⁻²⁷ It has the advantages of low cost, non-toxic, renewable, degradable and abundant reserves, which has some unique advantages comparing with commonly used polymers/binders. The hydroxyl group and amino group in CS structure make it has strong affinity, especially has good chelation ability for transition metals, coordinating with metal ions to form complexes, this property also provides a basis for dispersing metal active sites.²⁵⁻²⁸ In addition, CS has been proved to have the ability of structure guidance and good adsorption of CO₂.^{29, 30} These features of CS made it as interesting materials in designing electrocatalysts for CO₂RR.”. Please see **Page 2** and **3** in the revised manuscript.

Comment 3: A high EtOH selectivity at high current densities was achieved in a flow cell. Achieving record performance in membrane electrode assembly (MEA) would support the claim that this is a broadly applicable approach and one that advances the field toward practical application.

Response 3: We thank the reviewer for the comment. On the basis of the comment, we have conducted CO₂ electrolysis using MEA. The schematic diagram of MEA and the results were shown in **Figure S34 and S35**, respectively. As expected, it also achieved a high overall current of 1.2 A • cm⁻² with C₂₊ alcohols FE of 36.7% at -3.6 V cell voltage. The production rates of EtOH and PrOH could reach 1.54 mmol • h⁻¹ • cm⁻² and 0.50 mmol • h⁻¹ • cm⁻², respectively. In the revised manuscript, we have emphasized this by “We also carried out CO₂ electrolysis in membrane electrode assembly (MEA) (**Supplementary Figure S34 and S35**)^{36, 43, 44}. A high overall current of 1.2 A • cm⁻² with C₂₊ alcohols FE of 36.7% was achieved at -3.6 V cell voltage, and the production rates of EtOH and PrOH were 1.54 mmol • h⁻¹ • cm⁻² and 0.50 mmol • h⁻¹ • cm⁻², respectively.”. Please see **Page 11-12** in the revised manuscript. Accordingly, the specific experiment operations have also been supplemented in “CO₂ Electrolysis in membrane electrode assembly (MEA)” section of the revised manuscript. Please see **Page 20** in the revised manuscript.

Comment 4: This study was performed in an alkaline flow cell which leads to a severe carbonate formation, and the many associated issues documented well in this journal by the perspective by Kanan. How can the authors address the Kanan’s concerns?

Response 4: We thank the reviewer for the comment. Currently, KOH and KHCO₃ solution are both commonly used electrolytes for CO₂RR. We agree with the referee that performing in an alkaline flow cell leads to a severe carbonate formation, because we also observed this phenomenon in our experiment. Currently, intermittent cleaning is used to remove salt precipitation if long-term electrolysis is carried out. Alternatively, we think that CO₂ electrolysis in MEA is a promising approach for

practical application.

Comment 5: Most of the commercially available GDLs are hydrophobic. The reported contact angle in this study (95.1) is less than some previous reports. More explanation is needed to support why Cu-chitosan (CS)-GDL mitigates mass diffusion limitations and improves the CO₂ reduction performance at higher current densities.

Response 5: We thank the reviewer for the comment. Here we would like to discuss this briefly in response to the comment. Even though the commercially available GDLs are hydrophobic, we found that it is not the decisive factor to improve the catalytic activity. It has been found that controlling the electrode surface with appropriate contact angle was more conducive to form abundant gas-liquid-solid three-phase interface, which is favorable to improve the CO₂RR performance (*Nat Energy* **6**, 439-448 (2021); *Nat Catal* **18**, 1222-1227 (2019); *Nat Catal* **20**, 1000-1006 (2021); *Nat Catal* **1**, 592-600 (2018)) Therefore, we think the direct reason why 3D Cu-CS-GDL mitigates mass diffusion limitations and improves the CO₂ reduction performance at higher current densities was that when CS synergized with other components to form integrated 3D structure, it could form abundant gas-liquid-solid three-phase interface with more exposing active sites. This phenomenon leads to slightly reducing of the contact angle. To prove this, we determined the contact angle of a commercial hydrophobic carbon paper, Cu NPs-GDL, Cu/CS-GDL and De-Cu-GDL, which were 145.9°, 138.4°, 135.8° and 70° respectively (**Figure S30**). The lower contact angle of 3D Cu-CS-GDL than that of Cu NPs-GDL and Cu/CS-GDL indicates that the integrated 3D structure in 3D Cu-CS-GDL could form abundant gas-liquid-solid three-phase interface with more exposing active sites. However, a much lower contact angle of De-Cu-GDL leads to loss of gas diffusion ability. Accordingly, we have also determined electrochemical active surface areas (ECSA) using the reported method (*Nature* **529**, 68-71 (2016); *ACS Catal* **8**, 6560-6570 (2018)). The result shows clearly that ECSA values of 3D Cu-CS-GDL is larger than Cu NPs-GDL, indicating that more exposing active sites were formed in the abundant gas-liquid-solid three-phase interface. In addition, we have also added

EIS experiment to explain the enhancing electrons transport and mass diffusion of 3D Cu-CS-GDL, and the results support our conclusions.

In the revised manuscript, we have emphasized this by “This also can be known from the fact that when CS synergized with other components to form integrated 3D structure, it could form abundant gas-liquid-solid three-phase interface with more exposing active sites. This phenomenon leads to the lower contact angle of 3D Cu-CS-GDL (95.1°) than that of Cu NPs-GDL (138.4°) and Cu/CS-GDL (135.8°). However, a much lower contact angle of De-Cu-GDL (70°) leads to loss of gas diffusion ability (**Supplementary Figure S30**). The above result suggests that controlling the electrode surface with appropriate contact angle was more conducive to form abundant gas-liquid-solid three-phase interface with more exposing active sites, which is favorable to improve the CO₂RR performance.³⁸⁻⁴¹ As shown in **Supplementary Figure S31a-e**, the electrochemical active surface areas (ECSA) of the GDEs were estimated through the electrochemical double-layer capacitance (C_{dl}) measurements, and ECSA values of 3D Cu-CS-GDL, Cu NPs-GDL, De-Cu-GDL and Cu/CS-GDL were 600, 525, 360 and 125 cm²_{ECSA}, respectively.^{2, 42} After normalizing the current density to ECSA, 3D Cu-CS-GDL still exhibited the largest partial current densities of C₂₊ alcohols at the potential of -0.87 V vs RHE, which indicates that the 3D structure could improve the intrinsic activity for producing C₂₊ alcohols in CO₂RR (**Figure S31f**).”. Please see **Page 11** in the revised manuscript.

Comment 6: Cu NPs-GDL electrode shows a good performance at higher current densities (600-1000 mA/cm²). Is this related to the synthesis approach? A comparison between commercial Cu NP and the Cu NPs-GDL is needed.

Response 6: We thank the reviewer for the comment. We agree with the referee that the high performance is related to the synthesis approach. As suggested by the referee, we have also used commercial Cu NPs for comparison (**Figure S26**). The electrochemical CO₂RR performance was investigated (**Figure S27**). As a result, the FE of C₂₊ was only 10.3% and H₂ was the major product at 900 mA • cm⁻², indicating that the good performance of Cu NPs-GDL at high current density is related to the

synthesis approach. In the revised manuscript, we have added the discussion by “In addition, we have also used commercial Cu NPs for comparison (**Figure S26**). The real size of commercial Cu NPs was approximately 60 to 400 nm. As a result, the FE of C₂₊ was only 10.3% and H₂ was the major product at 900 mA •cm⁻² (**Figure S27**).”. Please see **Page 10** in the revised manuscript.

Comment 7: A deeper mechanistic investigation is needed to support the high EtOH FE. For example, Raman analysis can be compared to these works to further investigate the reaction pathways: J. Am. Chem. Soc. 2019, 141, 21, 8584, Nat Catal 3, 75–82 (2020), J. Mater. Chem. A, 2022, 10, 20059-20070.

Response 7: We thank the reviewer for the comment. According to the comment, in the revised manuscript, we have added more discussion on the Raman analysis “At 0.1 V vs RHE, except for 538 cm⁻¹, a peak can be observed at 364 cm⁻¹ in the Raman spectra of 344D Cu-CS-GDL electrode, corresponding to the restricted rotation of Cu–CO stretching. It was suggested that higher intensity of the Cu–CO stretching band can be assigned to a higher CO intermediate coverage and facilitate C–C coupling.⁴⁶ After -0.2 V vs RHE, it separated into two peaks located at 305 and 380 cm⁻¹, respectively, and corresponded to the Cu–CO frustrated rotation and the Cu–CO stretch, then disappeared at -0.7 V vs RHE.^{45, 47-49}” and “Otherwise, there was the C≡O stretching on Cu located about 2068 cm⁻¹ in Raman spectra of both 3D Cu-CS-GDL and Cu NPs-GDL, which can be deconvolved into top-bound CO and bridge-bound, suggesting that the pathway of generating C₂₊ products was in progress.⁵¹”. Please see **Page 12 and 13** in the revised manuscript. The related reference has also been cited (**Ref. 46, 48 and 51**).

Comment 8: XAS analysis shows some interesting results. I do recommend revisiting the results and combining them with the mechanistic study in the main text.

Response 8: We thank the reviewer for the comment. As suggest by the referee, in the revised manuscript, we have added the discussion by “In **Supplementary Figure S36**, the XAFS data of 3D Cu-CS-GDL are provided at OCV, -0.4 V, -0.8 V vs RHE during

CO₂RR and after reaction. The K-edge XANES spectra and the derivative K-edge XANES spectra indicated that 3D Cu-CS-GDL presented zero valence Cu in the whole process of CO₂RR. All curves in k space also followed the trend of the curve of Cu foil, which also proved that Cu (0) was maintained in CO₂RR. In R space, the ever-present Cu-Cu bond confirmed to the above conclusion, but its corresponding radial distance was shifted with applied potential, which is caused by surface adsorption and lattice vibration in the reaction environment. Therefore, the internal bonding of 3D Cu-CS-GDL was constant.”. Please see **Page 12** in the revised manuscript.

Comment 9: XRD needs to be revised, I suggest removing graphitic peaks and focus on the Cu peaks. If the interface of Cu (200) and Cu (111) is very important in EtOH pathway and production, this needs to be experimentally shown.

Response 9: We thank the reviewer for the comment. As suggested by the referee, we have removed the graphitic peaks and focus on the Cu peaks with the range from 30° to 60° . Please see **Figure 2g, S9, S13, S17 and S33** in the revised manuscript.

Comment 10: The results in this paper are impressive but do not exceed state-of-the-art performance to my understanding. E.g: Small Methods 2022, 6 (2), 2101334, Joule 2021, 5 (2), 429.

Response 10: We thank the reviewer for the comment. In the revised manuscript, we have changed the statement to “**which was very efficient for C₂₊ alcohols production**” and “**the as-synthesized 3D Cu-CS-GDL electrode was among the outstanding catalysts for C₂₊ products, especially for the high-rate production of C₂₊ alcohols (Figure 3d, Supplementary Table S1).**”. Please see **Page 1 and 10** in the revised manuscript. The related reference has also been cited (**Ref. 36, 37**).

Minor comments:

Comment 11: The authors claimed that “Unfortunately, construction of 3D nanostructured catalysts on porous gas diffusion layer (GDL) is very difficult”.

Spraying a mixture of a catalyst and ionomer (which is very common) leads to a 3D structure on GDL. Therefore, making a 3D structure is not challenging.

Response 11: We thank the reviewer for the comment. To avoid confusing, we have deleted the sentence and revised the statement by “While an electrodeposition process can also create a 3D material, it is difficult to deposit a 3D catalyst directly on a GDL due to the hydrophobic surface of the GDL”. Please see **Page 3** in the revised manuscript.

Comment 12: In some case, the total FE is higher than 100%, indicating an error.

Response 12: We thank the reviewer for the comment. In the manuscript, we have added error bar to estimate the FE in the product distribution. The total error of FE of eight products will be controlled to be $\pm 8\%$, which is among the reasonable range, similar to many reported works (*Nat Commun* **10**, 3851 (2019); *Angew Chem Int Ed* **134**, (2022); *Nat Commun* **13**, 3754 (2022); *J Am Chem Soc* **144**, 10446-10454 (2022)).

REVIEWERS' COMMENTS

Reviewer #2 (Remarks to the Author):

The authors have satisfactorily addressed the reviewer's comments.

Reviewer #3 (Remarks to the Author):

I appreciate the comprehensive revisions and responses here, and recommend publication.

Responses to the comments and revisions made

Reviewer #2 (Remarks to the Author):

Comments to the Author

The authors have satisfactorily addressed the reviewer's comments.

Response: We thank the reviewer very much for positive comment. Thank you for your careful interpretation of the article and your pertinent comments and suggestions, which make the article more rich in content layering, and more rigorous and scientific. Thank you again for your guidance.

Reviewer #3 (Remarks to the Author):

Comments to the Author

I appreciate the comprehensive revisions and responses here, and recommend publication.

Response: We thank the reviewer for the comment. We really appreciate your support and encouragement for our work. Because of your guidance and correction, the article is more innovative and scientific. In the process of replying to your comments, we have a more profound understanding and perception of the scientific issues in the article. Thank you again for your support.